

# Facing the infinity: tackling large samples of challenging Chironomidae (Diptera) with an integrative approach

Caroline Chimeno[1], Björn Rulik[2], Alessandro Manfrin[3], Gregor Kalinkat[4], Franz Hölker[4] and Viktor Baranov[5]

[1] Bavarian State Collection of Zoology (SNSB-ZSM), Munich, Germany
[2] Zoological Research Museum Alexander Koenig, Leibniz Institute for the Analysis of Biodiversity Change (LIB), Bonn, Germany
[3] Institute for Environmental Sciences, iES Landau, RPTU University of Kaiserslautern-Landau, Landau, Germany
[4] Leibniz Institute of Freshwater Ecology and Inland Fisheries (IGB), Berlin, Germany
[5] Estación Biológica de Doñana-CSIC/Doñana Biological Station-CSIC, Seville, Spain

## ABSTRACT

**Background:** Integrative taxonomy is becoming ever more significant in biodiversity research as scientists are tackling increasingly taxonomically challenging groups. Implementing a combined approach not only guarantees more accurate species identification, but also helps overcome limitations that each method presents when applied on its own. In this study, we present one application of integrative taxonomy for the highly abundant and particularly diverse fly taxon Chironomidae (Diptera). Although non-biting midges are key organisms in merolimnic systems, they are often cast aside in ecological surveys because they are very challenging to identify and extremely abundant.

**Methods:** Here, we demonstrate one way of applying integrative methods to tackle this highly diverse taxon. We present a three-level subsampling method to drastically reduce the workload of bulk sample processing, then apply morphological and molecular identification methods in parallel to evaluate species diversity and to examine inconsistencies across methods.

**Results:** Our results suggest that using our subsampling approach, identifying less than 10% of a sample's contents can reliably detect >90% of its diversity. However, despite reducing the processing workload drastically, the performance of our taxonomist was affected by mistakes, caused by large amounts of material. We conducted misidentifications for 9% of vouchers, which may not have been recovered had we not applied a second identification method. On the other hand, we were able to provide species information in cases where molecular methods could not, which was the case for 14% of vouchers. Therefore, we conclude that when wanting to implement non-biting midges into ecological frameworks, it is imperative to use an integrative approach.

Corresponding author
Caroline Chimeno,
chimeno@snsb.de

# INTRODUCTION

Chironomidae (non-biting midges) is by far the most ecomorphologically diverse and widely distributed ingroup of aquatic insects (*Hilsenhoff, Thorp & Covich, 2001*; *Armitage, Pinder & Cranston, 2012*). Occurring in every zoogeographic region, including Antarctica, non-biting midges inhabit nearly all aquatic and semiaquatic, marine and terrestrial habitats (*Armitage, Pinder & Cranston, 2012*). Characteristic behavioral and physiological adaptations have enabled these flies to colonize extreme environments such as caves up to 1,000 m deep, hot springs, high-altitude waters, glacial streams, and even highly polluted waters or sewage systems (*Andersen et al., 2016*; *Gadawski et al., 2022*). In aquatic systems, their abundance can be higher than that of all other macroinvertebrates combined, making them a keystone taxon in freshwater ecology (*Gratton & Zanden, 2009*; *Marziali et al., 2010*; *Karima, 2021*). The bottom-dwelling larvae not only represent almost every feeding group but, being ecosystem engineers, they also contribute enormously to sediment- and water-mixing, and to the global oxygen- and carbon-cycle (*Hölker et al., 2015*; *Baranov, Lewandowski & Krause, 2016*; *Antczak-Orlewska et al., 2021*). As ecosystem engineers, the Chironomidae are involved in modifying the availability of nutrients (chiefly phosphorous, but also nitrogen), as well as oxygen and carbon availability for other aquatic organisms (*Hölker et al., 2015*; *Baranov, Lewandowski & Krause, 2016*). All life stages (even the short-lived adults) play a vital role in aquatic and terrestrial food webs, serving as an important food source for fish, birds, bats and other arthropods (*Gratton & Zanden, 2009*; *Raunio, Heino & Paasivirta, 2011*; *Armitage, Pinder & Cranston, 2012*; *Wirta et al., 2015*; *Herren et al., 2017*). This combination of high ecosystem functionality, high abundance, and habitat specificity of the Chironomidae to their environment makes them suitable biological indicators for ecological assessments (*e.g.*, water quality control) (*Sæther, 1977*; *Lencioni, Marziali & Rossaro, 2012*; *Dorić et al., 2021*).

Despite this, only a limited subset of biodiversity studies or biomonitoring surveys of aquatic habitats incorporate species- or genus-level information of the Chironomidae and oftentimes, they are neglected altogether (*Raunio, Heino & Paasivirta, 2011*; *Dorić et al., 2021*). This is due to several factors: (i) non-biting midges are relatively difficult to identify (*Cranston, 2008*; *Proulx et al., 2013*), (ii) only few taxonomists with the required expertise are available for species-level identification (*Cranston et al., 2013*; *Chan et al., 2014*), (iii) traditional morphological-based species delimitations often require laborious dissection and mounting of specimens on microscope slides (*Ekrem, Stur & Hebert, 2010*; *Gadawski et al., 2022*), and (iv) they can be extremely species rich even in relative low-diversity temperate and boreal ecosystems (*Lundström et al., 2010*). The workload associated with the processing of non-biting midges from large bulk samples, common in ecological surveys, is immense when applying traditional identification methods (*Rosenberg, 1992*; *Brodin et al., 2012*). In humid climates, or during wetter years, the number of specimens to be processed can increase from hundreds of thousands to sometimes millions of specimens.

There are few methods that can help overcome the pitfall of processing an "infinite" number of specimens, with the most obvious one (and most resource-demanding) being

the employment of more taxonomists or parataxonomists (*Engel et al., 2021*) to help accelerate specimen processing and identification. The availability of expert taxonomists, however, is in decline and even then, financing such manpower at a large scale is often not feasible and remains time-consuming (*Hausmann et al., 2020*; *Chimeno et al., 2022*). Therefore, researchers often subsample bulk samples to reduce the sorting effort, or limit sample processing to a few key families or species (*Mandelik, Roll & Fleischer, 2010*; *Porter et al., 2014*; *Keck et al., 2017*; *Bohan et al., 2017*; *Chimeno et al., 2023*). One promising alternative that is currently in development is the use of automatic machine-based identification approaches for species identification (see *Milošević et al., 2020*). As demonstrated by Milošević and authors, after vigorously training their artificial neural network on 1,836 specimens belonging to ten similar-looking species of Chironomidae, they recovered 99% identification success when presenting their network new images. Despite these promising results, this technology is not yet applicable at a large scale because it requires laborious sample preparation and a vigorous training-phase of the target taxa (*Milošević et al., 2020*).

Currently, one of the most common and promising methodologies for large-scale species identification is DNA barcoding, a molecular-based identification method (*Brodin et al., 2012*; *Morinière et al., 2016*). It uses a short DNA fragment to differentiate species from one another, and does so at a lower cost and faster pace than traditional morphological methods (*Hebert et al., 2003*; *Ekrem, Willassen & Stur, 2007*; *Porter et al., 2014*; *Morinière et al., 2016*). With the rise of DNA barcoding, high quality species-level information of Chironomidae is increasingly becoming more accessible to research (*Ekrem, Stur & Hebert, 2010*; *Baloğlu, Clews & Meier, 2018*), and studies examining the efficiency of this method in research of these insects reveal an overall congruence of 80–90%, making it a great complement to taxonomic methodologies (*Carew, Pettigrove & Hoffmann, 2005*; *Pfenninger et al., 2007*; *Ekrem, Willassen & Stur, 2007*; *Carew et al., 2007*; *Carew, Marshall & Hoffmann, 2011*; *Lin, Stur & Ekrem, 2015*). However, just as any identification method, DNA barcoding has its own limitations (*Dayrat, 2005*; *Will, Mishler & Wheeler, 2005*; *Schlick-Steiner et al., 2010*) and therefore, numerous studies resort to applying a combined methodological approach for species identifications (*Pires & Marinoni, 2010*; *Sheth & Thaker, 2017*).

With many studies highlighting the need for a smart and efficient integration of both morphological and molecular species identification methods (*Hausmann et al., 2020*; *Hartop et al., 2022*), our study aims to present and evaluate one way to do so for a particularly diverse and complicated group of insects: the Chironomidae. To tackle the large amounts of insect material, we apply a three-level subsampling technique that we present in the Methods section. We also compare our DNA- and morphology-based species identifications in terms of accuracy, to demonstrate how the use of each method on its own can provide discrepant results. We are processing bulk samples of Diptera that have been collected in the framework of the federal-funded field experiment "Verlust der Nacht" (https://www.igb-berlin.de/projekt/verlust-der-nacht) and the follow-up project "Artenschutz durch umweltfreundliche Beleuchtung" (https://www.igb-berlin.de/projekt/artenschutz-durch-umweltvertraegliche-beleuchtung-aube) located in the Westhavelland

Nature Park in northeast Germany. The project was launched in 2012 with the goal of studying the effects that artificial lighting at night has on species communities.

## MATERIALS AND METHODS

### Study area and experimental design

The "Verlust der Nacht" experiment was conducted by the Leibniz Institute of Freshwater Ecology and Inland Fisheries (IGB) in a large-scale facility established in 2012 (see Holzhauer et al. (2015), Manfrin et al. (2017) for details). The facility is located in a 750-km$^2$ Dark-Sky Reserve within the Westhavelland Nature Park in the Berlin-Brandenburg Metropolitan Region (https://www.darksky.org/our-work/conservation/idsp/reserves/westhavelland/). The landscape is characterized by a system of drainage ditches (approximately 5 m wide, average annual water depth 50 ± 26 cm). In the grassland adjacent to the drainage ditch, we installed three parallel rows (3, 23 and 43 m away from the drainage ditch) of four conventional 4.75 m high streetlights located 20 m apart. Each lamp post in the lit site was equipped with one 70-W high-pressure sodium lamp (VIALOX NAV-T Super 4Y, yellow 2,000 K, Osram, Munich, Germany). In the control (dark) site only the lamp posts were installed (i.e., without bulbs) providing identical physical structure yet remaining dark. The lamps used in the lit site had a maximum illuminance of approximately 50 lx directly under the lamp, with the minimum illuminance between two adjacent streetlamps of the same row being approximately 10 lx, and a minimum illuminance between rows of streetlamps of ca. 1 lx (see Holzhauer et al. (2015) for further details about light distribution and spectral composition). From spring 2012 onward, the lit site was illuminated at night, i.e., between civil twilight at dusk and dawn. The lit and control sites are very similar in their environmental characteristics (e.g., water physico-chemistry, hydromorphology, riparian vegetation) and ~600 m (800 m along the drainage ditch) apart, separated by a row of trees.

### Insect collection

We collected insects emerging from the drainage ditch from both lit and dark sites from May to October 2014. Emerging insects were sampled using four floating pyramidal emergence traps (0.85 m × 0.85 m, 300-μm mesh), placed in the drainage ditch ca. 1 m from the bank and directly in front of each streetlamp. Sampling duration ranged from seven (one night samplings) to approximately 185 h (1 week samplings) and occurred monthly except in July when the sampling was conducted twice. Flying adult non-biting midges were collected from the grassland adjacent to the drainage ditch using 24 flight interception traps, 12 at each site. Flight intercepting traps were placed 0.5 m below each lamp and consisted of two perpendicular acrylic panels (each 204 mm × 500 mm × 3 mm) mounted above a collecting funnel. The flight intercepting traps were collecting insects for one 24-h sampling period every month except in July when sampling was conducted twice. Based on astronomical sunset and sunrise, the 24-h sampling periods were always split into a night-sampling (8–14 h, depending on the season) followed by a day-sampling (10–16 h), replacing the collecting jars after each of them. Sampling always occurred on rainless days/nights within 24 h of either first- or third-quarter moon. Both emergence and flight

intercepting traps were equipped with collecting jars containing 70% ethanol as a preservative medium (see *Manfrin et al. (2017)* for further details).

## Morphotype sorting and subsampling for processing

We obtained bulk samples of pre-sorted adult "Nematoceran" flies (crane flies, midges, gnats, mosquitoes *etc.*) stored in 90% ethanol that were collected in the sampling year 2014 (see "Insect collection"). From these samples, our senior author, who is a trained expert of non-biting midges, sorted specimens using a stereo microscope and grouped them into different morphotypes. To do this, we used three different approaches based on the "difficulty" of specimen sorting (Fig. 1). Large and/or conspicuous species that are easy to recognize, such as *Prodiamesa olivacea* (*Meigen, 1818*) or *Ablabesmyia phatta* (*Egger, 1863*), were quickly sorted into their own distinct morphotypes and assigned a preliminary species name. Specimens that were more difficult to group (because they belong to genera that have similar-looking representatives when viewed under the stereo microscope) were sorted at the genus-level, hence, grouped into genera-morphotypes if possible. Hence, if several genera have similar-looking representatives under the stereo microscope, we sorted representatives of several genera into one morphotype. Lastly, for specimens that our expert taxonomist found difficult to address, subsets were mounted on temporary glycerol slides to be examined at ×400 magnification in a first step, so that similar specimens can be assigned to the same morphotype in a second step. From every morphotype group, we selected a representative number of morphotype voucher specimens (about 10%). For very abundant morphotypes where 10% of specimens is still too much, we sampled fewer individuals. Selected specimens were used for molecular and morphological species identifications.

## Sequencing of selected specimens

For specimens larger than 2 mm, we used a single leg or leg segment as a tissue sample that was transferred to a 96-well plate. For smaller individuals, we extracted DNA non-destructively (*i.e.*, subsequent voucher recovery) from the whole body. After lysis, we extracted genomic DNA using the BioSprint96 magnetic bead extractor and the respective kits by Qiagen (Hilden, Germany). We carried out a polymerase chain reaction (PCR) in a total reaction volume of 20 µl, including 2 µl of undiluted DNA template, 0.8 µl of each primer (10 pmol/µl), 2 µl of 'Q-Solution' and 10 µl of 'Multiplex PCR Master Mix', containing hot start Taq DNA polymerase and buffers. The latter components are available in the Multiplex PCR kit by Qiagen (Hilden, Germany).

Thermal cycling was performed on GeneAmp PCR System 2,700 machines (Life Technologies, Carlsbad, CA, USA) as follows: hot start *Taq* activation: 15 min at 95 °C; first cycle set (15 repeats): 35 s denaturation at 94 °C, 90 s annealing at 55 °C (−1 °C/cycle) and 90 s extension at 72 °C. Second cycle set (25 repeats): 35 s denaturation at 94 °C, 90 s annealing at 40 °C and 90 s extension at 72 °C; final elongation 10 min at 72 °C. As established within the German Barcode of Life (GBOL) project at the ZFMK, we used the standard degenerate barcoding primers LCO1490-JJ: 5′-CHACWAAYCATAAAGATATYGG- 3′ and HCO2198-JJ: 5′-AWACTTCVGGRTGVCCAAARAATCA- 3′ (*Astrin & Stüben, 2008*).

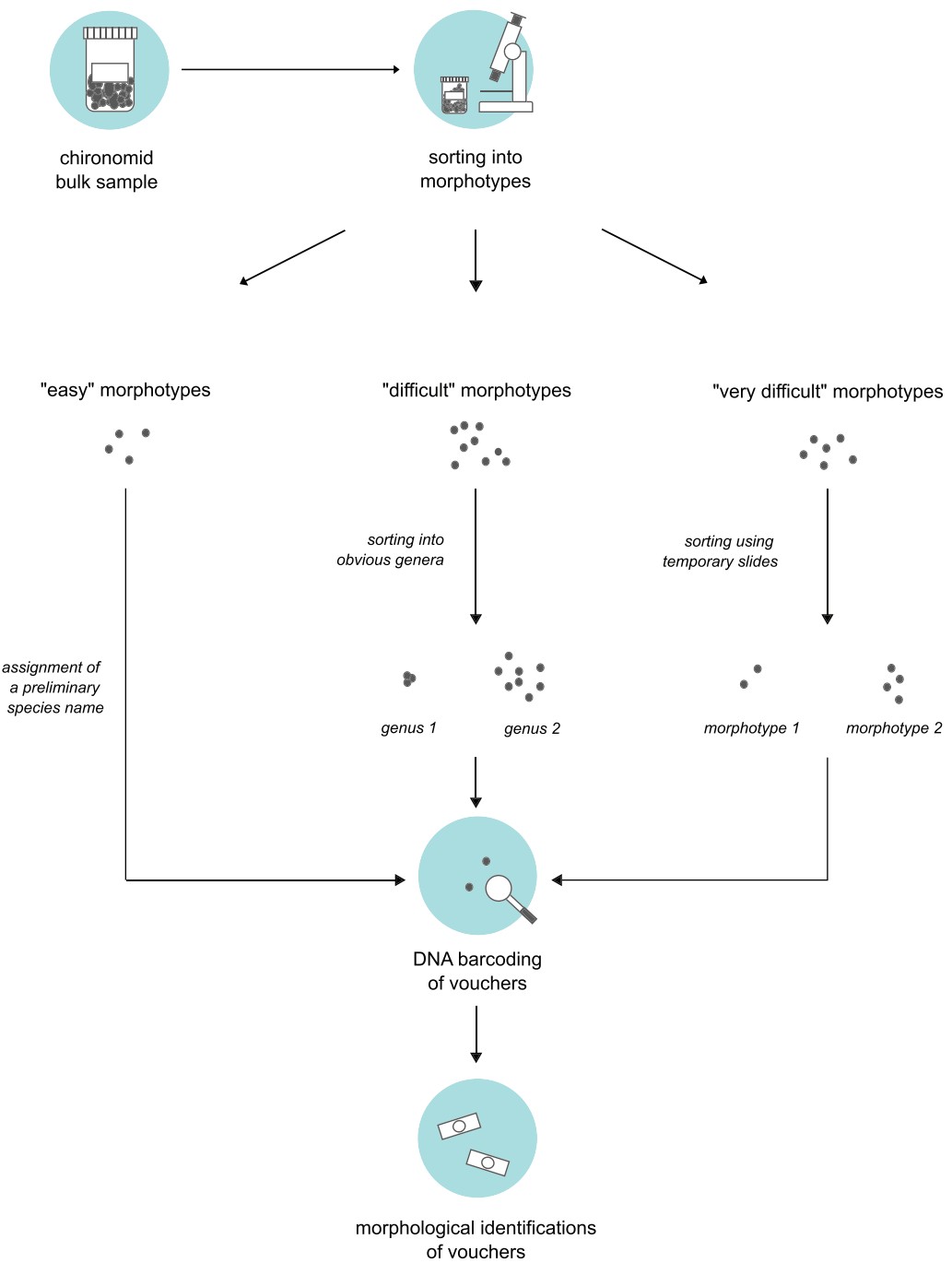

**Figure 1** **Three-level sorting workflow that was used in this study for bulk sample processing.** For each morphotype distinguished in a bulk sample, we conducted morphological & molecular identifications of selected vouchers. The procedure was different based on the difficulty of the specimens involved in sorting.

Purification and sequencing were conducted by the BGI Group (Hong Kong, China) using the amplification primers.

Traces were semi-automatically edited, then assembled sequences using the MUSCLE alignment approach (*Edgar, 2004*), and checked for the occurrence of stop-codons or hints

of nuclear mitochondrial DNA segments (NUMTs) in Geneious version 7.1.9 (http://www.geneious.com; *Kearse et al., 2012*). Further details such as voucher information, primer pairs, sequence data and trace files were deposited to BOLD and GenBank. These can be found under the following information (http://doi.org/10.5883/DS-ALANCHIR; GenBank accession numbers OP927392–OP927685).

## Morphological identifications

After DNA barcoding (or in parallel, depending on whether whole specimens were transferred to plates or just tissue samples), we mounted the specimens (or their empty shells) on permanent slides in Euparal and Hydromatrix following standard procedure (*Kirk-Spriggs & Sinclair, 2017*). Morphological identifications were conducted with aid of numerous identification keys and papers covering palaearctic Chironomidae (see *Lehmann (1970)*, *Saether (1971)*, *Hirvenoja (1973)*, *Wiederholm (1989)*, *Ekrem (2002a)*, *Langton & Pinder (2007)*, *Pillot (2008)*, *Giłka (2011)*). These identifications were conducted by our senior author which has conducted various research on the taxonomy of Chironomidae (see *Baranov (2011a)*, *2011b)*, *2013)*, *Baranov & Perkovsky (2013)*, *Baranov & Przhiboro (2014)*, *Baranov, Andersen & Hagenlund (2015)*, *Baranov, Andersen & Perkovsky (2015)*, *Baranov, Góral & Ross (2017)*, *Baranov et al. (2019)*).

## DATA ANALYSIS

All sequence records including metadata were uploaded to the online database Barcode of Life Data System (BOLD; *Ratnasingham & Hebert, 2007*). Sequences ≥300 base pairs (bp) were automatically assigned a Barcode Index Number (BIN) on BOLD if sequence similarity based on the (RESL-) BIN algorithm was fulfilled. Sequences ≥500 bp which did not find a match served as founders of new BINs. The dataset was downloaded on April 11, 2022, for analysis and can be viewed on Figshare (https://doi.org/10.6084/m9.figshare.21803013). Therefore, the present results correspond to BINs assigned at that time (BIN assignments can change as new sequences are added to BOLD). In addition to using the RESL-algorithm that is implemented into BOLD, we also applied Assemble Species by Automatic Partitioning (ASAP; *Puillandre, Brouillet & Achaz, 2021*) and SpeciesIdentifier version 1.9 (*Meier et al., 2006*) to cluster our sequences at 3%. ASAP uses pairwise genetic distances for hierarchical clustering without using information on intraspecific diversity, and SpeciesIdentifier is an algorithm that allows to cluster sequences based on their pairwise intra- and interspecific genetic distances. The outputs of all three algorithms were used to compare the number of Operational Taxonomic Units (OTUs) obtained with each and comparing diversity assessments. To compare all methodologies, we created a Neighbor-Joining in MEGA11 (version 11.0.13) of all sequence data and added morphological species-, ASAP-, RESL-, and SpeciesIdentifier labels (Data S1). Because all depict similar performance (see results), subsequent taxonomic analyses were conducted only using the RESL outputs.

To assess our sampling effort, we calculated Chao1 and Chao2 estimates using the *ChaoSpecies* function of the *SpadeR* package (version 0.1.1; *Chao et al., 2016*) in R (version 4.2.1) on abundance and incidence data, respectively (Data S2). We did this to estimate the
species diversity at the sampling site and to compare it to that which was empirically observed in our samples. Then, we used the *iNEXT* function from the *iNEXT* package (version 3.0.0; *Hsieh, Ma & Chao, 2016*; *Hsieh, Ma & Chao, 2020*) to extrapolate the species diversity obtained with each methodology (morphology, RESL, ASAP, and SpeciesIdentifier) to double the sampling effort. To depict the species diversity recovered per morphotype, we created accumulation curves using the *iNEXT* function on results derived from each identification method (morphological and molecular).

To double-check our identifications and to recover possible misidentifications, we created a dataset from BOLD containing 19,525 public COI-sequences of 1,035 species of non-biting midges collected throughout Europe (Data S3). We applied the following selection criteria to build a neighbor-joining tree: Kimura 2 Parameter distance model, sequences ≥200 bp, and excluding contaminants, records flagged with stop codons, and records flagged as misidentifications. To facilitate review, we colored the tree based on barcode clusters (BINs). We added the names of identifiers along with the identification method to each entry to discriminate high-level taxonomists that used morphological methods to vouchers from parataxonomists relying on the BOLD engine for sequence identification. We considered expert identifications as those conducted by researchers with taxonomic experience of Chironomidae, such as Elisabeth Stur (Norwegian University of Science and Technology; Norway; see *Stur & Ekrem (2000*, *2006*, *2011*, *2015)*, *Stur & Wiedenbrug (2005)*, *Stur & Spies (2011))*, Torbjørn Ekrem (Norwegian University of Science and Technology, Norway; see *Ekrem (2002a*, *2002b*, *2007)*, *Ekrem & Stur (2009)*, *Ekrem, Stur & Hebert (2010))*, Yngve Brodin (Swedish Museum of Natural History, Sweden; see *Brodin, Lundström & Paasivirta (2008)*, *Siri & Brodin (2014))*, Piotr Gadawski (University of Lodz; Poland; see *Gadawski et al. (2022)*, *Giłka & Gadawski (2022)*,)), and Sophie Wiedenbrug (University of São Paulo, Brazil) (see *Wiedenbrug, Lamas & Trivinho-Strixino (2012*, *2013)*, *Silva & Wiedenbrug (2015)*, *Wiedenbrug & Silva (2016))*.

## RESULTS

### Identification of specimens

Overall, we sorted through 4,549 specimens of non-biting midges which made up the bulk (99.6%) of "Nematoceran" specimens in our samples. We recovered 48 morphotype groups, and in total selected 331 specimen-vouchers, of which more than half were females (Data S2).

### *Molecular identifications*

We applied DNA barcoding to all 331 specimens and obtained 315 COI-barcodes (95%) that we uploaded to BOLD. Five sequences contained cross contaminations, and another 16 were identified as not being non-biting midges, but species of the taxa Anisopodidae, Chaoboridae, Culicidae, Hybotidae, Psychodidae, Sciaridae, and Trichoceridae.

The remaining COI-sequences were clustered into 77 BINs which provided coverage for 55 species and four interim species (essentially being morphotype analogs that are widely used in ecological studies) (*Ablabesmyia sp. 2ES*, *Smittia sp. 8ES*, *Smittia sp. 14ES*, and *Thienemanniella sp. 3TE*). Interim species names are assigned on BOLD when molecular

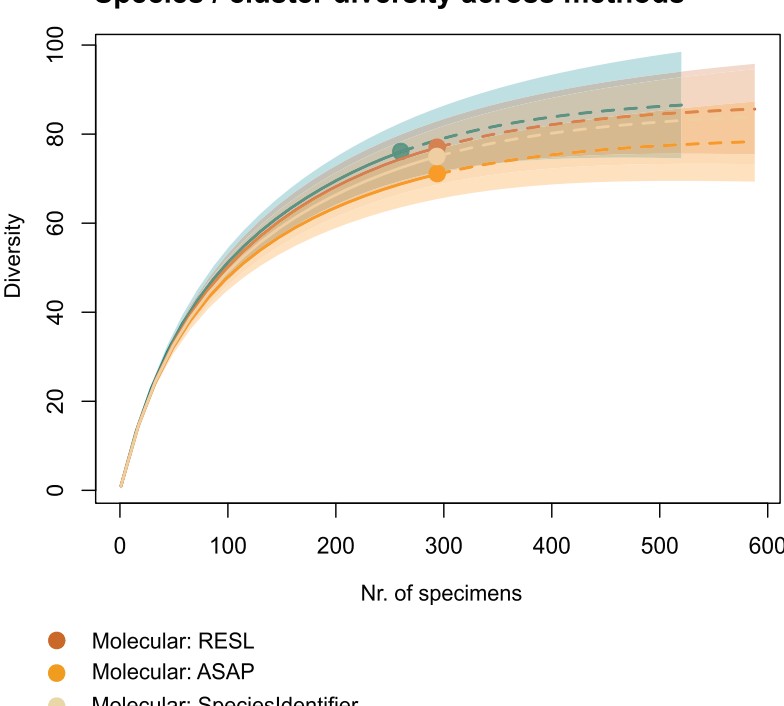

**Species / cluster diversity across methods**

● Molecular: RESL
● Molecular: ASAP
● Molecular: SpeciesIdentifier
● Morphology

**Figure 2 Accumulation curves of species and clusters recovered across methods.** Dotted lines represent extrapolated values (for up to double the sampling effort), bold lines represent interpolated values. Accumulation curves show the number of morphologically identified species and that of clusters recovered with RESL, ASAP, and SpeciesIdentifier.

analysis detects genetic differences, but no species name can be provided due to the lack of a taxonomic revision or of formal species description (*Stur & Ekrem, 2011*; *Morinière et al., 2016*). Seven BINs did not provide conclusive species-level identification and five BINs did not match to public data, providing no molecular identification. In five cases, two BINs were assigned to the same species (*Cladopelma viridulum*—BOLD:AAD7363 and BOLD: AAV3586; *Polypedilum cultellatum*—BOLD:AAH7761 and BOLD:ACX5929; *Polypedilum sordens*—BOLD:ACY3855 and BOLD:ADF3485; *Smittia stercoraria*—BOLD:AAN5358 and BOLD:AAN5355; *Smittia sp. 14ES* —BOLD:AAM7064 and BOLD:ACW5117). Data S2 provides an overview of the entire dataset.

We applied two other clustering algorithms (SpeciesIdentifier and ASAP) to our COI data. Although both SpeciesIdentifier (using 3% threshold) and ASAP (1st partition) did suggest slightly fewer clusters than the RESL-algorithm, all derived species diversities fall into the 95% confidence interval (Fig. 2), and the results were largely consistent across methods (Tables 1–3, Figs. 3B–3C).

### Morphological identifications

Using morphological methods, we identified a total of 76 species. A total of 34 specimens were left unidentified at a higher taxonomic level: 22 at the genus-, and 12 at the family-level.

**Table 1 Chao1/2 estimates and iNEXT extrapolation values across methods.**

| Method/Algorithm | Output | Values |
|---|---|---|
| **Morphology** | Sample size (n) | 260 |
| | Number of tax. entities | 76 |
| | Number of rare entities | 44 |
| | Sample coverage | 0.91 |
| | Chao1 estimate | 89 ± 7 SE |
| | iNEXT extrapolation (2n) | 87 ± 12 SE |
| | Chao2 estimate | 109 ± 15 SE |
| | Jackknife SE/bias | 0.0036/0 |
| **Molecular: RESL** | Sample size (n) | 294 |
| | Number of clusters | 77 |
| | Number of rare clusters | 40 |
| | Sample coverage | 0.93 |
| | Chao1 estimate | 87 ± 6 SE |
| | iNEXT extrapolation (2n) | 86 ± 10 SE |
| | Chao2 estimate | 100 ± 11 SE |
| | Jackknife SE/bias | $0.0039/-2.3502e^{-14}$ |
| **Molecular: ASAP** | Sample size (n) | 294 |
| | Number of clusters | 71 |
| | Number of rare clusters | 34 |
| | Sample coverage | 0.94 |
| | Chao1 estimate | 79 ± 5 SE |
| | iNEXT extrapolation (2n) | 78 ± 9 SE |
| | Chao2 estimate | 92 ± 11 |
| | Jackknife SE/bias | 0.0042/0 |
| **Molecular: SpeciesIdentifier** | Sample size (n) | 294 |
| | Number of clusters | 75 |
| | Number of rare clusters | 39 |
| | Sample coverage | 0.93 |
| | Chao1 estimate | 85 ± 6 SE |
| | iNEXT extrapolation (2n) | 84 ± 11 SE |
| | Chao2 estimate | 98 ± 11 |
| | Jackknife SE/bias | 0.0040/0 |

**Note:**
Results after applying Chao1 and Chao2 biodiversity calculations to each datatype (morphological; molecular: RESL, ASAP, SpeciesIdentifier), including sample sizes (Nr. of specimens), taxonomical entities (Nr. of species for morphological data; clusters for molecular data), sample coverage, Chao1 and Chao2 estimates, jackknife validations, and extrapolations to double the sample size.

## Assessing our sampling effort

Chao1 species richness estimates suggest that 79 ± 5 to 89 ± 7 species may have been present in the community that we sampled (Table 1). Sample-based Chao2 estimates were slightly higher, suggesting 92 ± 11 to 109 ± 15 species. Extrapolation to double the sampling effort would have increased the number of recovered entities by 11–17% (Fig. 2). Sample coverage was above 90% for all data (morphology, RESL, ASAP, SpeciesIdentifier).

**Table 2 Cases of discrepancies between morphological and molecular-based identifications.**

| Discrepancy | Morphotype | Nr. of sequences | Morphological ID of specimen | BIN | Molecular ID linked to BIN |
|---|---|---|---|---|---|
| **Type 1** | "Acricotopus lucens" | 2 | Acricotopus lucens | BOLD:AAG5487 | Procladius crassinervis |
| | "Chironomus" | 1 | Chironomus plumosus | BOLD:ACT6966 | Chironomus obtusidens |
| | "Chironomus" | 1 | Chironomus prasinatus | BOLD:AAU4046 | Chironomus annularius |
| | "Chironomus" | 1 | Chironomus sp. | BOLD:ADF1214 | Benthalia carbonaria |
| | "Dicrotendipes" | 1 | Dicrotendipes tritomus | BOLD:AAU1021 | Dicrotendipes nervosus |
| | "Endochironomus" | 2 | Endochironomus albipennis | BOLD:AAW5643 | Endochironomus tendens |
| | "Endochironomus" | 1 | Endochironomus stackelbergi | BOLD:AAW5643 | Endochironomus tendens |
| | "Glyptotendipes" | 1 | Glyptotendipes cauliginellus | BOLD:ACD4470 | Glyptotendipes pallens |
| | "Glyptotendipes" | 1 | Glyptotendipes glaucus | BOLD:ACD4470 | Glyptotendipes pallens |
| | "Glyptotendipes" | 1 | Glyptotendipes glaucus | BOLD:AAC0597 | Glyptotendipes paripes |
| | "Parachironomus" | 3 | Parachironomus gracilior | BOLD:ACY5073 | Parachironomus monochromus |
| | "Paratanytarsus/Rheotanytarsus" | 1 | Paratanytarsus laetipes | BOLD:AAI6018 | Cricotopus bicinctus |
| | "Procladius ferrugineus" | 2 | Procladius ferrugineus | BOLD:AAG5487 | Procladius crassinervis |
| | "Procladius pectinatus" | 1 | Procladius pectinatus | BOLD:ACW5385 | Procladius culiciformis |
| | "Pseudosmittia obtusa" | 1 | Pseudosmittia obtusa | BOLD:ACP4407 | Pseudosmittia trilobata |
| | "Smittia aterrima" | 2 | Smittia aterrima | BOLD:AAN5358 | Smittia stercoraria |
| | "Tanypus punctipennis" | 1 | Tanypus punctipennis | BOLD:ADJ7832 | Tanypus kraatzi |
| | "Tanytarsus" | 1 | Tanytarsus reei | BOLD:ACF7553 | Tanytarsus heusdensis |
| | "Tanytarsus" | 2 | Tanytarsus dispar | BOLD:ACG9929 | Tanytarsus medius |
| | "Xenopelopia nigricans" | 1 | Xenopelopia nigricans | BOLD:ADJ7832 | Tanypus kraatzi |
| **Type 2** | "Ablabesmyia phatta" | 1 | Ablabesmyia phatta | BOLD:ACK3818 | Ablabesmyia sp. 2ES |
| | "Chironomidae" | 12 | Chironomidae sp. | BOLD:AAC0597 | Glyptotendipes paripes |
| | "Cladopelma/Cryptochironomus/Harnischia" | 1 | Cladopelma sp. | BOLD:AAV3586 | Cladopelma viridulum |
| | "Cladopelma/Cryptochironomus/Harnischia" | 1 | Cladopelma sp. | BOLD:AAV8096 | Cladopelma virescens |
| | "Endochironomus" | 9 | Endochironomus sp. | BOLD:AAW5643 | Endochironomus tendens |
| | "Glyptotendipes" | 1 | Glyptotendipes sp. | BOLD:ACD4470 | Glyptotendipes pallenses |
| | "Psectrocladius" | 1 | Psectrocladius sp. | BOLD:AAU0273 | Psectrocladius limbatellus |
| | "Smittia terrestris" | 2 | Smittia terrestris | BOLD:ACP4736 | Interim species Smittia sp. 8ES |
| | "Smittia terrestris" | 7 | Smittia terrestris | BOLD:ACW5117 | Interim species Smittia sp. 14ES |
| | "Thienemanniella" | 1 | Thienemanniella vittata | BOLD:AAV3048 | Interim species Thienemanniella sp. 3TE |
| **Type 3** | "Acricotopus lucens" | 1 | Acricotopus lucens | BOLD:AEO5089 | No public data |
| | "Chironomus" | 6 | Chironomus curabilis | BOLD:ACD8415 | Chironomus curabilis/nuditarsis |
| | "Cricotopus" | 2 | Cricotopus sp. | BOLD:AEO5089 | No public data |
| | "Cricotopus" | 3 | Cricotopus sylvestris | BOLD:AAA5299 | Cricotopus sylvestris/glacialis |
| | "Cricotopus" | 1 | Cricotopus tricinctus | BOLD:AEG4456 | Cricotopus tricinctus/sylvestris/trifasciatus |

(Continued)

| Discrepancy | Morphotype | Nr. of sequences | Morphological ID of specimen | BIN | Molecular ID linked to BIN |
|---|---|---|---|---|---|
| | "*Cricotopus*" | 9 | *Cricotopus sylvestris* | BOLD:AAA5299 | *Cricotopus sylvestris/glacialis* |
| | "*Glyptotendipes*" | 4 | *Glyptotendipes cauliginellus* | BOLD:AAF8348 | *Glyptotendipes cauliginellus/ lobiferus* |
| | "*Metriocnemus*" | 1 | *Metriocnemus sp.* | BOLD:ADV3586 | No public data |
| | "*Microtendipes chloris*" | 5 | *Microtendipes chloris* | BOLD:ACY5270 | *Microtendipes pedellus/chloris* |
| | "*Parachironomus*" | 1 | *Parachironomus sp.* | BOLD:ADV3586 | No public data |
| | "*Procladius crassinervis*" | 4 | *Procladius crassinervis* | BOLD:ACB6320 | *Procladius sp.* |
| | "*Psectrocladius oxyura*" | 1 | *Psectrocladius oxyura* | BOLD:AEO4348 | No public data |
| | "*Tanytarsus usmaensis*" | 2 | *Tanytarsus usmaensis* | BOLD:AEO0788 | No public data |

**Note:**
Morphotypes, number of sequences, and identifications that were involved in discrepant results, namely complete incongruences in identification across methods (type 1), molecular methods provided more species-level information than morphology (type 2), and Morphology provided more species-level information while molecular methods provided inconclusive or no identification at all (type 3).

## Discrepancies between morphology- and DNA-based identifications

Overall, we recovered discrepant identifications among 103 specimens (Table 2), and categorized them as follows:

Type 1: Cases with complete incongruence in identifications across methods (27 specimens).

Type 2: Molecular methods provided higher taxonomic resolution than morphology (36 specimens).

Type 3: Morphology provided higher taxonomic resolution while molecular methods provided inconclusive or no identification at all (40 specimens).

Meticulous revision of our molecular and morphological data revealed that all type-1 discrepancies were caused by misidentifications that were performed by the senior author (Viktor Baranov), which involves 9% of all voucher specimens. For another 9% of vouchers, morphological identifications could not provide identifications at the species-level (type-2), meaning that for a total of 18% of vouchers, morphology did not provide accurate or comprehensive species-level identifications.

On the other hand, morphological identification methods did provide more comprehensive species information for a total of 40 specimens (14%). Here, we were able to provide species-level IDs for five BINs that did not provide public data on BOLD, and for six BINs that were linked to discrepant identifications by taxonomists.

## Uncovering species diversity from morphotypes

Of the 48 morphotypes that we distinguished during sorting, we identified 77 species (including misidentifications) using morphology and 78 BINs using molecular methods (Table 3). The most abundant (and thus higher sampled) morphotypes within our samples were "MT Glyptotendipes", "MT Parachironomus", "MT Paratanytarsus/Rheotanytarsus", "MT Cladopelma/Cryptochironomus/Harnischia", and "MT Cricotopus". These morphotypes encompass 42% (125) of all analyzed specimens. Species identification,

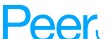

**Table 3 Overview of all analysed specimens of Chironomidae.** Number of specimens, morphologically identified species, BINs, ASAP- and SpeciesIdentifier OTUs recovered per morphotype.

| Morphotype | Specimens | Morph. identified species | BINs | ASAP | SP-ID |
|---|---|---|---|---|---|
| "Ablabesmyia longistyla" | 1 | 1 | 1 | 1 | 1 |
| "Ablabesmyia monilis" | 2 | 1 | 1 | 1 | 1 |
| "Ablabesmyia phatta" | 2 | 1 | 1 | 1 | 1 |
| **"Acricotopus lucens"** | **3** | **1** | **2** | **2** | **2** |
| "Benthalia" | 1 | 1 | 1 | 1 | 1 |
| "Chironomidae" | 12 | 0 | 1 | 1 | 1 |
| **"Chironomus"** | **14** | **5** | **6** | **5** | **6** |
| "Cladopelma/Cryptochironomus/Harnischia" | 22 | 5 | 6 | 5 | 6 |
| "Coryneura" | 6 | 2 | 2 | 2 | 2 |
| "Cricotopus" | 20 | 4 | 5 | 5 | 5 |
| **"Dicrotendipes"** | **5** | **2** | **1** | **1** | **1** |
| **"Endochironomus"** | **12** | **2** | **1** | **1** | **1** |
| **"Glyptotendipes"** | **32** | **5** | **4** | **4** | **4** |
| "Guttipelopia guttipennis" | 11 | 1 | 1 | 1 | 1 |
| "Kiefferulus tendipediformis" | 4 | 1 | 1 | 1 | 1 |
| "Metriocnemus atriclava" | 1 | 1 | 1 | 1 | 1 |
| "Metriocnemus" | 2 | 1 | 1 | 1 | 1 |
| "Microchironomus" | 5 | 1 | 1 | 1 | 1 |
| "Microtendipes chloris" | 5 | 1 | 1 | 1 | 1 |
| "Microtendipes pedellus" | 1 | 1 | 1 | 1 | 1 |
| "Nanocladius dichromus" | 1 | 1 | 1 | 1 | 1 |
| "Orthocladius oblidens" | 2 | 1 | 1 | 1 | 1 |
| **"Parachironomus"** | **26** | **4** | **5** | **5** | **5** |
| "Paraphaenocladius impensus" | 2 | 1 | 1 | 1 | 1 |
| **"Paratanytarsus/Rheotanytarsus"** | **25** | **6** | **7** | **6** | **7** |
| "Polypedilum sordens" | 6 | 1 | 2 | 2 | 2 |
| "Polypedilum" | 10 | 2 | 3 | 2 | 3 |
| "Procladius crassinervis" | 10 | 1 | 2 | 2 | 2 |
| "Procladius culiciformis" | 5 | 1 | 1 | 1 | 1 |
| **"Procladius ferrugineus"** | **2** | **1** | **1** | **1** | **1** |
| "Procladius nigriventris" | 2 | 1 | 1 | 1 | 1 |
| **"Procladius pectinatus"** | **1** | **1** | **1** | **1** | **1** |
| "Procladius" | 3 | 0 | 1 | 1 | 1 |
| "Psectrocladius limbatellus" | 5 | 1 | 2* | 1 | 1 |
| "Psectrocladius oxyura" | 2 | 1 | 2* | 2 | 2 |
| "Psectrocladius" | 1 | 0 | 1 | 1 | 1 |
| "Pseudosmittia albipennis" | 1 | 1 | 1 | 1 | 1 |
| **"Pseudosmittia obtusa"** | **1** | **1** | **1** | **1** | **1** |
| **"Smittia aterrima"** | **2** | **1** | **1** | **1** | **1** |
| "Smittia edwardsi" | 2 | 1 | 1 | 1 | 1 |
| "Smittia stercoraria" | 1 | 1 | 1 | 1 | 1 |

(Continued)

| Table 3 (continued) | | | | | |
|---|---|---|---|---|---|
| Morphotype | Specimens | Morph. identified species | BINs | ASAP | SP-ID |
| "Smittia terrestris" | 9 | 1 | 3* | 2 | 2 |
| **"Tanypus punctipennis"** | **1** | **1** | **1** | **1** | **1** |
| "Tanypus vilipennis" | 1 | 1 | 1 | 1 | 1 |
| "Tanytarsus usmaensis" | 2 | 1 | 1 | 1 | 1 |
| **"Tanytarsus"** | **8** | **6** | **4** | **4** | **4** |
| "Thienemanniella" | 1 | 1 | 1 | 1 | 1 |
| **"Xenopelopia nigricans"** | **1** | **1** | **1** | **1** | **1** |
| **Total** | **294** | **76** | **88** | **71** | **75** |

Notes:
Morphotype-names are in quotation marks, and those that include morphological misidentifications are in bold.
* Includes multiple BINs.

revealed that each of these morphotypes comprise 4–7 different taxonomic entities. In 15 cases, more BINs than morphologically identified species were recovered per morphotype. Morphotypes that include morphological misidentifications are in bold.

We created accumulation curves based on our morphological (Fig. 3A) and molecular data (Fig. 3B), depicting the number of recovered taxonomic entities for the most diverse morphotypes (with at least four taxonomic entities), and extrapolating to double the sampling effort. Most morphotypes that depict an accumulation curve, reach an asymptote. Comparing graphs, we see that in some cases, too many species were identified morphologically per morphotype (see "MT Tanytarsus" and "MT Glyptotendipes") and too few in others (see "MT Paratanytarsus/Rheotanytarsus").

## DISCUSSION

In this study, we applied an integrative approach to facilitate sample processing of highly diverse non-biting midges. We applied a three-level subsampling technique and compared species recovered with each identification method (molecular and morphological) with the goal of assessing how an integrative approach can increase the incorporation of the Chironomidae into monitoring programs and biodiversity studies using a simplified approach (but without losing too much species information).

### Morphotype sorting

Our results suggest that our morphotype sorting method was successful: We obtained a coverage of over 90% in species and cluster counts (Table 1), and the plateauing accumulation curves in Fig. 2 indicate that we would not have captured substantially more species by increasing our sampling effort. This is interesting, because after sorting non-biting midges into morphotype groups, we ultimately processed and identified only 7% of all specimens. Considering this, we believe that the task of grouping them into morphotypes, then selecting specimens for subsequent analysis can be easily delegated to parataxonomists. Overall, in-depth knowledge of Chironomidae morphology is not essential for this stage of sample processing, because sorting is based on phenotypic traits

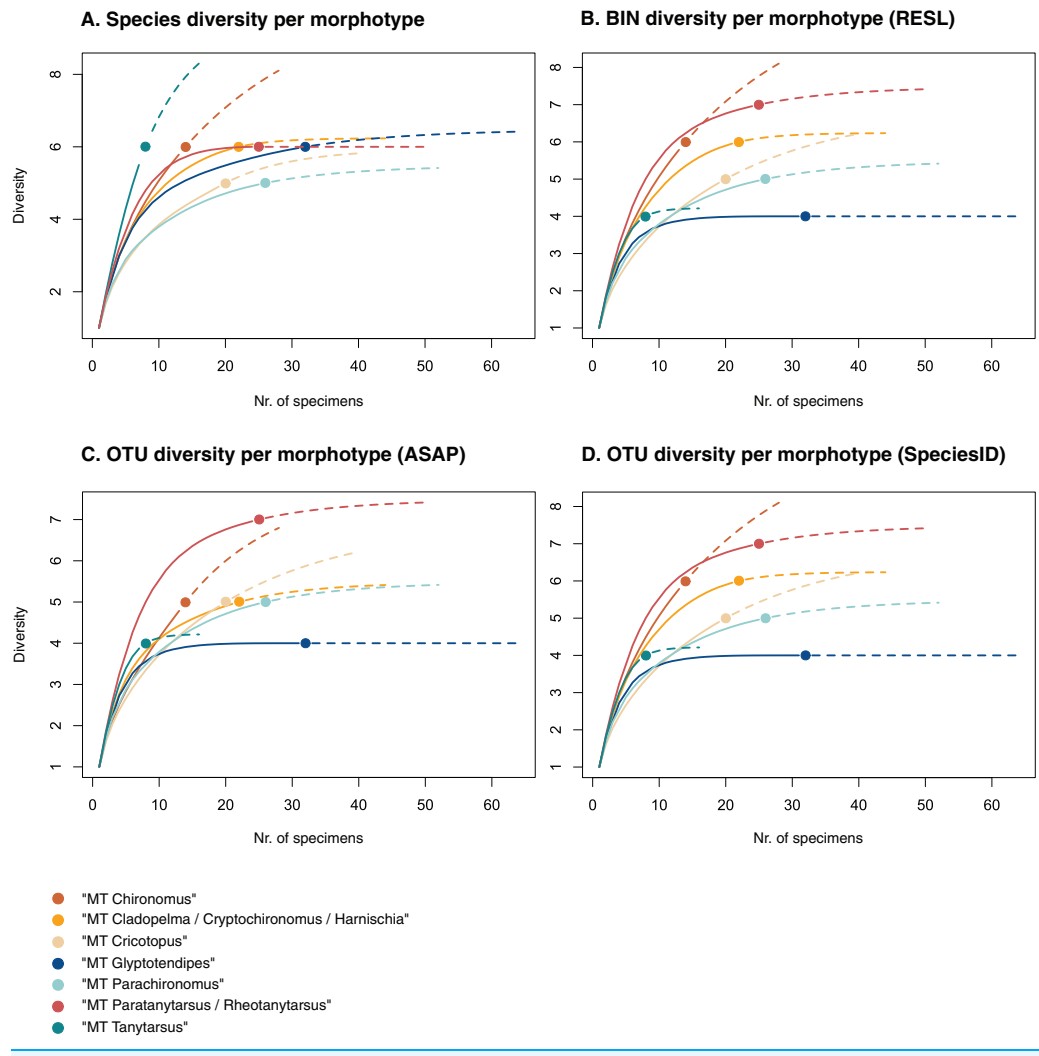

**Figure 3 Accumulation curves of the diversity of (A) species, (B) BINs, (C) ASAP-OTUs, and (D) SpeciesIdentifier-OTUs recovered for each chironomid morphotype.** Dotted lines represent extrapolated values (up to double the sampling effort), bold lines represent interpolated values. Accumulation curve of number of morphologically identified species (A) and BINs (B) recovered per morphotype based on the number of sampled specimens. Multiple BIN cases have been accounted for and removed.

such as size, coloration, venation, setation, and shapes of antennae which simply require having a good "eye" and patience (*Krell, 2004*; *Ekrem, Stur & Hebert, 2010*). This approach was also applied by *Ekrem, Stur & Hebert (2010)* and authors to subsample non-biting midges for analysis in their study. We are aware that in our case, sorting was not conducted by a parataxonomist, but by an experienced scientist (*Ekrem, Stur & Hebert, 2010*). However, our taxonomist sorted these directly from the ethanol fluid using a stereo microscope, which does not provide a high-enough resolution for distinguishing genus- or species-level morphological features, especially not in ethanol. When confronted with large numbers of especially challenging specimens, our taxonomist resorted to either mounting representatives on temporary slides for guidance, or grouping specimens in the very few

genera that have distinct features even at low resolutions (*e.g.*, *Cricotopus*, *Ablabesmyia* and *Tanypus*). Our identifications of voucher specimens recovered up to seven taxonomic entities per single morphotype, indicating that when in doubt, it is simply easier to merge more specimens into one larger morphotype and compensate by increasing the number of vouchers.

Applying Chao statistics, we estimated that about 80 putative species may be present at the sampling sites. However, it is important to mention that to a certain degree, we are still underestimating the actual diversity of the Chironomidae that are present at the sampling sites. We applied our Chao statistics to a subset of the data, meaning that we are unintentionally inflating the probability of encountering a "new" and/or rare species, which in turn results in lower species estimates. To counteract this, we additionally applied a sample-based Chao2 estimator on the incidence data, which, resulted in much higher species estimates (Table 1). Needless to say, we may still be underestimating species numbers.

## Using DNA barcoding: working with species proxies

In our study, we clustered our COI sequences using three delimitation algorithms, namely RESL, ASAP, and SpeciesIdentifier. Because the RESL algorithm and its BIN system is directly integrated into BOLD's interface, it is commonly used in DNA barcoding applications. However, there are varying opinions regarding the sole use of BINs for species delimitation (see *Cranston et al., 2013*; *Meier et al., 2022*), especially when assuming that BIN numbers are equal to species numbers in a 1:1 ratio. Therefore, as recommended by *Cranston et al. (2013)*, we analyzed our sequence data with several delimitation methods that apply different clustering algorithms. It is important to note that regardless which method one chooses for analysis, clustering algorithms remain arbitrary. Our results indicate that all three algorithms performed well, with molecular operational taxonomic unit (MOTU) diversities derived from each depicting overlapping 95% confidence intervals. Overall, we obtained very comparable results for all three clustering methods. In fact, using the NJ-tree to depict the assignment of specimens into clusters depicted almost identical results (see Data S1).

Using the RESL-algorithm led to the assignment to 77 BINs. Although BINs are a strong proxy for species boundaries (*Zahiri et al., 2014*; *Hebert et al., 2016*), it is important to keep in mind that they do not always reflect existing taxonomic systems (*Raupach et al., 2010*; *Hausmann et al., 2013*; *Zahiri et al., 2014*; *Hawlitschek et al., 2017*). Incongruences between BINs and traditional species names include multiple BIN assignments (more than one BIN is detected in a traditionally recognized species) and BIN sharing (the same BIN is detected across more than one recognized species) (*Hawlitschek et al., 2017*; *Chimeno et al., 2022*). Ideally, multiple BIN assignments would imply the presence of cryptic diversity whereas BIN sharing, which is commonly found among taxa with uncertain taxonomy or challenging species groups, is an indication for the need of species synonymization (*Hausmann et al., 2013*). However, ideal conditions are not the rule and there are various molecular factors (such as heteroplasmy, numts sequencing, introgression or homogenization of mtDNA haplotypes) that can challenge COI-based species

identifications (*Kmiec, Woloszynska & Janska, 2006*; *Dobson, 2004*; *Pamilo, Viljakainen & Vihavainen, 2007*; *Duron et al., 2008*; *Buhay, 2009*; *Hazkani-Covo, Zeller & Martin, 2010*), making it important to incorporate morphological information whenever possible. Additionally, accurate species identification is only guaranteed provided that high quality reference libraries are being used as a backbone to analysis (*Ekrem, Willassen & Stur, 2007*; *Chimeno et al., 2019*). These, in turn, rely on the accuracy of morphological identifications conducted on voucher specimens (*Ekrem, Willassen & Stur, 2007*). Mistakes in reference databases are challenging to uncover, especially if one is working with molecular data only. Yet requesting taxonomists to meticulously revise identifications of vouchers is not feasible. Instead, we suggest that it is mandatory that all records uploaded to BOLD are provided with an identifier and identification method, so that others can rely on the data when no expert is available. As suggested by *Brodin et al. (2012)* and authors, reference databases need to be expanded as best as possible in order to provide a better taxonomic coverage of species and their intraspecific variation. Quantity, however, should not come at a cost of quality. In our case, we double-checked every molecular-based identification using a neighbor-joining tree of public sequence data of vouchers that were morphologically identified by a taxonomist and uploaded to BOLD. Sequence records that were either identified using the "BIN taxonomy match" tool on BOLD, or that did not provide any information on the method of voucher identification whatsoever, were disregarded completely.

Discordances in our molecular dataset include multiple BINs assignments for a total of seven species, and the assignment of four interim species names. Although multiple BIN-assignments are an indication for cryptic diversity, extensive analysis is required to uncover the driving factors in the recovered genetic differences. On the other hand, interim species names are assigned to BINs when a genetic difference is detected, yet no species name can be provided. This can be an indication for the need of a taxonomic revision or a formal species description (*Morinière et al., 2019*; *Ekrem et al., 2019*). In other words: Interim species names provide species with an "intermediate name" until they obtain a formal species name. Because of this, such species can still be implemented into analyses, as in our study, because their BIN assignments act as "taxonomic handles" (see *Morinière et al. (2016)*, *Geiger et al. (2016)*).

The seven species involved in multiple-BIN cases are *Cladopelma viridulum*, *Polypedilum cultellatum*, *Polypedilum sordens*, *Psectrocladius oxyura*, *Psectrocladius limbatellus*, *Smittia stercoraria*, and *Smittia terrestris*. Research has shown that these genera (especially *Cladopelma*, *Polypedilum*, *Pscetrocladius* and *Smittia*) display much higher intraspecific variations in the COI barcode region across species, making it hard to identify a barcode gap that is needed for species discrimination (*Pillot, 2008*; *Cranston, Hardy & Morse, 2012*; *Tang et al., 2022*). These genera include species complexes whose taxonomic position is yet unsolved, and many traditional species are suspected to comprise more than one cryptic diverse species that are awaiting formal description (*Lehmann, 1970*; *Saether, 1971*; *Carew, Pettigrove & Hoffmann, 2005*; *Song et al., 2018*; *Chimeno et al., 2022*). *Song et al. (2018)*, for example, recovered a total of five BINs for *P. cultelatum* without finding any morphological discrepancies between adult specimens, and therefore

concluded that they may be dealing with potential cryptic species within a species complex. However, when *Carew, Pettigrove & Hoffmann (2005)* did not find DNA marker-associated morphological variations among individuals of the genus *Cladopelma*, they realized that this was due to the fact that these variations are only present among immature stages.

With the increase in barcoding campaigns, more COI-data of the Chironomidae is being made publicly available. One valuable asset of DNA barcoding is the fact that different life stages of the same species can be easily linked to one another without having to undergo larvae rearing which can be time-consuming, expensive, and for some species very challenging (*Stoeckle, 2003*; *Blaxter, 2004*; *Ekrem, Willassen & Stur, 2007*; *Stur & Ekrem, 2011*). With increased sequencing of larval stages, the COI sequences can be matched with those inferred from adult species and thus help enormously in resolving at least some taxonomic uncertainties (*Carew, Pettigrove & Hoffmann, 2005*; *Sinclair & Gresens, 2008*; *Montagna et al., 2016*).

### Using morphology for species delimitation

In contrast to molecular identification methods, which use an algorithm for unbiased taxonomic clustering, accurate morphological identifications rely highly on (1) the availability and accuracy of species determination keys and (2) the identifier's ability to conduct identifications from an objective perspective (*Ekrem et al., 2019*). Chironomid identification requires extensive knowledge (which can generally only be provided by an expert) and ideally, as demonstrated by *Carew, Pettigrove & Hoffmann (2005)*, more than one single life-stage (*e.g.*, adults) of a single species should be assessed. Unfortunately, taxonomic expertise is overall in steady decline especially for those working on small-bodied and less conspicuous taxa (*Engel et al., 2021*; *Chimeno et al., 2022*). Still, the availability of a taxonomist does not automatically guarantee error-free species identifications, as demonstrated in this and other studies (*Failla et al., 2016*). Not only did we have a 9% error rate among morphological identifications, six of the "single species morphotypes" that were said to be distinguishable enough under the stereo microscope for direct species assignment were incorrectly identified. For another 9% of specimens, we could only provide identification to the family or to the genus-level.

False identifications were almost always within a given genus, hence, between closely related species whose morphological differences are often very subtle and therefore require specimen mounting and meticulous analysis (*Ekrem, Stur & Hebert, 2010*). For diverse morphotypes, the number of taxonomic entities recovered using morphology was often over- or under-estimated. This reflects the fact that on one hand, these taxa can display high levels of intraspecific morphological variation (*Carew et al., 2007*; *Carew, Marshall & Hoffmann, 2011*), and on the other hand, closely related species exhibit strong similarities, leading to the erroneous synonymization of species (*Anderson, Stur & Ekrem, 2013*). Despite having drastically reduced our taxonomist's workload by analyzing only a small portion of collected individuals, our taxonomist still spent about 500 active working hours processing, mounting, and identifying specimens, which was prone to errors over time (person. comment Baranov). This is a stark contrast to the 63 working hours for our

molecular approach. Although females are known to be even more difficult to identify than males, misidentifications were much more frequent among male individuals (70% of all type-1 discrepancies).

Overall, despite applying a three-level subsampling approach, which reduced the processing workload drastically, the performance of our taxonomist was affected by mistakes, caused by large amounts of material. These large amounts of material, however, represent the everyday life conditions in ecological surveys. For almost 20% of selected vouchers, no species-level information was provided, and we therefore conclude that it is difficult to meet the requirements of ecological studies using morphology alone.

## CONCLUSION

Our current contribution shows that while both morphological identification and DNA barcoding have their own limitations, they are highly complementary in tackling large insect samples. While DNA barcoding does not require difficult-to-acquire taxonomic knowledge and drastically fast-forwards the process of identification of non-biting midges, barcode registries are only as valuable as the quality of their vouchers. Hence, without morphological identifications, there is no DNA barcoding. We presented one way to apply an integrative approach on Chironomidae, and presented a three-level sorting method for large samples. We were able to demonstrate that DNA barcoding less than 10% of a sample's contents can reliably detect >90% of its diversity, bringing us one step closer towards optimizing processing workflows for very large insect samples.

## ACKNOWLEDGEMENTS

We are very grateful to the German Barcode of Life initiative (GBOL II) for providing the opportunity of specimen sequencing. We thank Martin Spies for the help with voucher verification, and Jörg Lewandowski (IGB Berlin) for supporting Viktor Baranov during his PhD, which allowed for side projects (such as this one) to take place.

### Funding

Verlust der Nacht has been financially supported by the German Federal Ministry of Education and Research (BMBF-033L038A). Sequencing was done in the framework of the German Barcode of Life initiative (GBOL II), which was also funded by the German Federal Ministry of Education and Research (BMBF-01LI1101, BMBF-01LI1501). Alessandro Manfrin was funded by the Deutsche Forschungsgemeinschaft (DFG, German Research Foundation)–326210499/GRK2360. Viktor Baranov received funds supporting the payment of the PeerJ publication fees through the CSIC Open Access Publication Support Initiative from the CSIC Unit of Information Resources for Research (URICI). Viktor Baranov's work is sponsored by the Spanish State Agency for Innovation's Ramon y Cajal fellowship (RyC2021-032144-I), project title "Climate change in the past and present & Insect decline". The funders had no role in study design, data collection and analysis, decision to publish, or preparation of the manuscript.

## Grant Disclosures

The following grant information was disclosed by the authors:

German Federal Ministry of Education and Research: BMBF-033L038A, BMBF-01LI1101, BMBF-01LI1501.

German Barcode of Life initiative (GBOL II).

Deutsche Forschungsgemeinschaft (DFG, German Research Foundation): 326210499/GRK2360.

CSIC Unit of Information Resources for Research (URICI).

Spanish State Agency for Innovation's Ramon y Cajal Fellowship: RyC2021-032144-I.

## Competing Interests

The authors declare that they have no competing interests.

## Author Contributions

- Caroline Chimeno analyzed the data, prepared figures and/or tables, authored or reviewed drafts of the article, and approved the final draft.
- Björn Rulik performed the experiments, authored or reviewed drafts of the article, and approved the final draft.
- Alessandro Manfrin conceived and designed the experiments, authored or reviewed drafts of the article, and approved the final draft.
- Gregor Kalinkat conceived and designed the experiments, authored or reviewed drafts of the article, and approved the final draft.
- Franz Hölker conceived and designed the experiments, authored or reviewed drafts of the article, and approved the final draft.
- Viktor Baranov performed the experiments, authored or reviewed drafts of the article, and approved the final draft.

## DNA Deposition

The following information was supplied regarding the deposition of DNA sequences:

All 294 sequences are available at GenBank: OP927392–OP927685 and BOLD https://dx.doi.org/10.5883/DS-ALANCHIR.

The data spreadsheets downloaded in April 2022 from BOLD (including sequencing and metadata) are available at Figshare: Chimeno, Caroline (2023): BOLD Spreadsheets. figshare. Dataset. https://doi.org/10.6084/m9.figshare.21803013.v1.

## Data Availability

The data spreadsheets downloaded in April 2022 from BOLD (including sequencing and metadata) are available at Figshare: Chimeno, Caroline (2023): BOLD Spreadsheets. figshare. Dataset. https://doi.org/10.6084/m9.figshare.21803013.v1.

The raw code for R analysis is available at Figshare: Chimeno, Caroline (2023): R Code for "Facing the Infinity". figshare. Software. https://doi.org/10.6084/m9.figshare.21787259.v3.

The input dataset for the R code is available at Figshare: Chimeno, Caroline (2023): Input Dataset for R code. figshare. Dataset. https://doi.org/10.6084/m9.figshare.21787262.v2.

## Supplemental Information

Supplemental information for this article can be found online at http://dx.doi.org/10.7717/peerj.15336#supplemental-information.

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
