# Peer review of "Facing the infinity: tackling large samples of challenging Chironomidae (Diptera) with an integrative approach"

_PeerJ, doi:10.7717/peerj.15336_

## Round 0.1 · original submission · Major Revisions

This is an interesting submission and an approach that would make identifying Chironomidae in aquatic samples less challenging. Both reviews had positive things to say about the current submission. However, there are some comments and suggestions from both that need to be addressed, where possible. Personal preferences of the reviewers aside, consider the valid points made and revise according. Provide valid rationales and justifications for the choices made for variables, indices, and approach to the study.

Reviewer 1 ·

Basic reporting

1. Basic reporting
This manuscript mostly complies with baseline criteria. It is well-structured, well-set against the scientific background, has good figures and all raw data are accessible. Concerning language, the authors seem to use jargon in some sections (e.g., when referring to specimens as being «notoriously difficult» to identify), and I believe that Figure 3 could be prepared using a bit different symbols (this is just a personal comment, I dislike the R basic plotting symbols) and maybe nicer colours. I don't understand the colour-coding scheme for the rarefaction plots (Figure 3): there seem to be 4 colours, where 3 would have been appropriate. I don't quite understand Figure 2 and I also have no means of changing that – is this figure correctly labelled and what is this figure meant to display? And is it correctly represented in the text?
Also, I am not quite sure whether the Hill numbers are correctly represented by the authors. I don't know the package, but Hill's index at q=1 is the exponential of the Shannon index, and the reciprocal of the Simpson index at q=2 (e.g., Magurran & McGill 2011). Not, as the authors allude, the Shannon or the Simpson diversity. This must be corrected and properly assessed.
It is not clear whence the samples actually came from – the section Insect collection provides an overview over the sampling procedure, but I cannot guess which of these samples (emergence traps, flight interception traps) actually went into the present work.

Experimental design

2. Experimental design
The experimental design is of high quality and produces good and useful data; methods are standard and adequately described. There are some aspects on the experimental design addressed in sect. 4, General comments. I challenge the use of BINs following the example of Meier et al. 2022.

Validity of the findings

3. Validity of the findings
Findings (i.e. that barcoding approaches can offer to assess biodiversity in a challenging group of insects) are valid, some aspects may want some reconsiderations (see sect. 4, General comments).

Additional comments

Review Chimeno et al. PeerJ #81108

The authors present a case-study that uses a subsampling approach to assess diversity in a highly diverse and taxonomically challenging group of aquatic insects, the Chironomidae. In addition to their high biodiversity, Chironomidae are key constituents of freshwater communities. This makes them excellent model for integrative taxonomy.
The authors collected insects through traps, presorted the bulk to obtain Chironomidae, and used what I consider a stratified subsampling strategy to economically process the samples by defining morphotypes at 3 different levels of «identifyablity» and obtain biodiversity estimates through standard and barcoding approaches.
In total, they used 331 specimens of a 4549 specimen bulk sample for their biodiversity assessment, representing 48 hypothetical taxa. They recover a quite congruent assessment of Chironomidae biodiversity, with some 100 specimens for which no conclusive identification could be achieved (which the authors differentiate into 3 groups: complete incongruence, molecular identification only, morphological identification only).
Also, they estimate total biodiversity in their sample based on the subsample using various species richness estimators, including Chao1.
Based on their results they argue that subsampling large samples comprising a high number of Chironomidae specimens by screening for distinct morphotypes is a strategy to estimate biodiversity.

This is a difficult manuscript to review. On the one hand, the authors present an interesting case and offer a strategy how to deal with large samples comprising a high number of specimens from a challenging group – on the other, there are some aspects that I find not well addressed, and some mistakes that should be corrected.

0. Custom checks
I was able to find, download and review the data. Well done.

1. Basic reporting
This manuscript mostly complies with baseline criteria. It is well-structured, well-set against the scientific background, has good figures and all raw data are accessible. Concerning language, the authors seem to use jargon in some sections (e.g., when referring to specimens as being «notoriously difficult» to identify), and I believe that Figure 3 could be prepared using a bit different symbols (this is just a personal comment, I dislike the R basic plotting symbols) and maybe nicer colours. I don't understand the colour-coding scheme for the rarefaction plots (Figure 3): there seem to be 4 colours, where 3 would have been appropriate. I don't quite understand Figure 2 and I also have no means of changing that – is this figure correctly labelled and what is this figure meant to display? And is it correctly represented in the text?
Also, I am not quite sure whether the Hill numbers are correctly represented by the authors. I don't know the package, but Hill's index at q=1 is the exponential of the Shannon index, and the reciprocal of the Simpson index at q=2 (e.g., Magurran & McGill 2011). Not, as the authors allude, the Shannon or the Simpson diversity. This must be corrected and properly assessed.
It is not clear whence the samples actually came from – the section Insect collection provides an overview over the sampling procedure, but I cannot guess which of these samples (emergence traps, flight interception traps) actually went into the present work.

2. Experimental design
The experimental design is of high quality and produces good and useful data; methods are standard and adequately described. There are some aspects on the experimental design addressed in sect. 4, General comments. I challenge the use of BINs following the example of Meier et al. 2022.

3. Validity of the findings
Findings (i.e. that barcoding approaches can offer to assess biodiversity in a challenging group of insects) are valid, some aspects may want some reconsiderations (see sect. 4, General comments).

4. General comments
Comments en gros

One aspect of my critique is that they ignore is the taxonomic quality of BOLD as regards the BIN system, and the how BOLD matches query sequences against the database.
I would be interested to learn whether the authors used the first hit or checked what the first 20 (random number here) tell them. In my experience (that does not cover Chironomids well) there sometimes some case where BOLD returns a best match to a query sequence despite there being several other sequences – also of different taxa – with similar similarities. This is, for instance, also acknowledged by dedicated software (e.g. Buchner et al 2020, BOLDigger current versions). It would be interesting to see if and how the taxonomic assignment and the biodiversity estimates change when also incorporating these results.
Their effort to segregate reference sequences into reliable ones and unreliable ones based on who identified the specimen behind the sequence is commendable, and arguably should become a standard procedure for taxonomic assignments through BOLD.

Also, I find myself wondering whether a species richness estimator such as Chao1 can be applied to the dataset the authors elaborated. There was an active selection process involved in defining which samples/specimens should go into barcoding and in my opinion this disagrees with the prerequisite of randomly subsampling a representative sample of an assemblage/community to achieve individual based rarefaction. I suspect that this means that the probabilistic considerations underlying rarefaction are violated to some degree, because the assumptions about abundance distributions et cetera are not correctly represented. For instance, Chao1 and Chao2 assume that an asymptote is reached when all species in the sample are represented by at least 2 individuals – and I am not sure how well the authors' subsampling scheme complies with that assumption: they state that they aimed to process about 10% of all specimens of all but the most abundant morphotypes. So, in this instance, the probability of encountering a «new» species was artificially inflated by excluding a large number of specimens of a few(?) very abundant taxa.
Another aspect that comes to mind here is that since the abundances the authors had probably did not properly represent abundance distributions in their sample, the use of the abundance-based Chao1 may not be entirely in order – maybe the authors can find a way to jackknife their samples (maybe possibly repeatedly so?) and apply the sample-based Chao2.

The authors highlight how laborious the standard approach is – 500 active working hours – but I would be interested in learning how that compares to the total efforts of the molecular approach. There will have been subsampling (unless performed by the taxonomist), entering data in datasheets, DNA extraction, DNA quantification, PCR amplification, checking of PCR success and PCR product concentrations, preparation of samples for sequencing (possibly by normalisation through dilution), editing and assembling traces (which can be quite laborious – I recently spent >20 h in editing and assembling a small but challenging dataset, but is not referred to in the manuscript), arranging data, preparing trees, matching query sequences to BOLD, extracting these results, and many more steps involved in this process. Of course, handling of samples at BGI, purification and sequencing also require human labor. So, I wonder how much time and effort was spent to assemble the molecular data.

Finally and most importantly, I challenge the sole use of BINs as surrogates for species. I would much rather see an analysis combining several! “species-delimitation” methods (such as e.g., ASAP, PTP, or even GMYC), the BIN-based assessment and the morphological analyses (see e.g., Meier et al. 2022 for a discussion). In my opinion, only then would this contribution qualify as integrative assessment of Chironomidae diversity. Especially for this group of insects a good and stable diversity assessment is necessary and these must be obtained through rigorous and consistent methods. I also challenge the assignment of interim nomina nuda species names – there is ample evidence that mtCOI is a poor proxy for species diversity, and that only a combination of nuclear and mitochondrial data that are analysed by appropriate means can serve to assess species diversity using molecular data. There also is the GAGE species concept, that negates the relevance of mitCOI data altogether (Seifert 2020). Therefore, I think that a significant adaptation of the molecular species delimitation methodology is in order.

There appear to be mismatches between text and abstract regarding the proportion of misidentifications.


Comments en detail (Numbers refer to line numbers)
a. Taxonomic comments: Italics throughout for species and genus names, also in graphs. When referring to morphotypes it would probably be more correct to clearly highlight them as such – e.g. by adding some morphotype identifyier ("MT Polypedilum" would be an option). Correct spelling of ALL misspelled taxon names – e.g. “Harschia” in the graphs, “Polypedium“ in the text etc.
b. Text
36:38 – I do not see why there should be an explanation for the misidentifications made by the last author. Humans err, taxonomists are human, ergo, taxonomists err. Unless, of course, the last author claims he could have identified all the material correctly had he not had to process a large number of specimens. Also, while it is interesting to learn that about 20% of all vouchers were misidentified, as reader I am wondering here whether there are cases where molecular based identification failed and why this is not highlighted in the abstract.
53 – I agree that aquatic insects are important parts of any aquatic ecosystem (and terrestrial ones for that matter), but I wonder if the term «ecosystem engineers» is correctly applied here. Is there evidence that chironomids alter aquatic ecosystems so that other taxa find a more suitable habitat? I am aware that this quality has been identified in e.g., beavers, but I would not readily call an aquatic insect an ecosystem engineer.
67 – confluence? Find more appropriate term.
87 – see Comments en gros
101:102 - one could also refer to the works of Pierre Taberlet et al. here
118:119 – no need for using a list [ (i) … (ii)]
173 – there is no section "Sampling techniques"
183 – notoriously? This sentence does not work – if specimens were notoriously difficult to identify they'd be known and there would be a community talking about these specimens. That's an easy fix. Maybe something in the lines of «that our expert taxonomist found difficult to address»? Note: rephrase/edit throughout.
186 – rephrase/split sentence to reduce complexity.
191:192 – repetition from above
211:217 – Did you get finalized sequences from BGI? Or was there some editing/assembly that was not reported?
238 – Is Chao1 appropriate?
243:245 – This description of the Hill numbers is not correct (see above), and again, I wonder whether the relative abundances the authors have are representative of their sample.s
246 – rather: … quantify richness in samples/plots.
277:278 – and what where these?
306 – What is "more species-level information"?
308 – Ditto. Do you mean «higher taxonomic resolution»?
311 – delete «morphological»
310:319 – I don't see how these figures match to what is reported in the abstract ("We conducted misidentifications for almost 20% of vouchers, which may not have been recovered had we not applied a second identification").
324:325 – Why are some morphotypes in italics and others not?
352:355 – I think there are better references for the use of parataxonomists/parataxonomist methods, but I would ask the authors to find a way to preserve this list of characters.
369:379 – there is much debate about what BINs are and what BINs are not (e.g., Meier et al. 2022), and I would welcome if the authors could stay clear off that - ideally by using a set of proper “species-delimitation” tools – but also by limiting this section to the bare necessities.
368:427 – I am afraid to state that I find this part of the discussion rather weak. In my opinion, it builds too strongly on the BOLD-proprietary BIN system and neglects the scope of following an integrative approach to study Chironomidae diversity.
401:411 – I don't see how and why this is relevant to the manuscript. Interim species names are nomina nuda and of little value. In the worst case, these will complicate future taxonomic work and make revisions necessary.
413:421 – I don't agree with this assessment. Wiemers & Fiedler 2007 showed that the barcoding gap can be the result of insufficient sampling, and unless these possibly widespread species are comprehensively sampled I would urge the authors to remain cautious in their interpretations of results.
461 – As stated above, to be human is to err, and as taxonomists invariably will be humans, they are going to err. In contrast, to blindly trust a database and a single algorithm is a fallacy.
463 – first correct use of "notoriously" in the manuscript

Conclusion: I suggest including proper species delimitation methods, and rewrite parts of the manuscript. I commend the authors on putting this dataset together and especially the work of the last author who carried to significant burden of identifying the specimens. However, the sole use of BINs as proxies for diversity appears to be inadequate, and this weakness of the manuscript should be resolved. The present state sells the whole work short: this is a beautiful dataset, and it deserves to be presented well. I understand that this is part of a PhD-thesis, but the analyses I suggest can be rapidly implemented (ASAP [https://bioinfo.mnhn.fr/abi/public/asap/] and PTP [https://species.h-its.org/] through webmasks, like GMYC – for reconstructing a tree I suggest using the IQ-tree interface which is available as webGUI [http://iqtree.cibiv.univie.ac.at/]); adding these results to (a potentially revisited) rarefaction of diversity is trivial. What may become a bit more challenging is bringing morphology and several molecular results together. What possibly reads like a major effort can likely achieved over the course of 3 weeks of consistent work.
Based on this assessment, I rule a major revision.

·

Basic reporting

The base idea of the paper addresses a relevant problem and provides a possible approach how to deal with it. I have only minor suggestions that should be considered.

Some of the comments will be well known to the authors, but I mention that in every review I do.

26 "family": modern phylogenetic framework is rankless. There are no criteria when something is a family (or sub-family), therefore ranks are arbitrary. I suggest to drop this expression from the entire manuscript.

27 "Chironomidae" is used here as a group (apparently). Important for further below.

27 "chironomid" is an ambiguous derivation. Why not simply use "non-biting midges": Check throughout text.

27 I am not sure whether there is a "merolimnic system". In my view, "merolimnic" is a character of a species or organism.

31 Here "Chironomidae" is used a plural form. As lain out in editorials in Systematic Entomology, authors should restrict their use either to address the group or its representatives. I suggest to stick to the first use here (the group).

43 "most diverse" concerning what? Quite some groups of merolimnic insects are more species-rich (Trichoptera, Odonatoptera).

272 As the authors have indicated above, "nematoceran" is not a valid reference; it can therefore not be used without any marking here.

321 I am not sure what "Linnean" species should indicate. Reference as a rank or a specific species concept? Also used further below.

324ff/335ff Are these genera names? Should these be in italics?

406 not sure whether "confirmed" is right, for philosophical reasons

408 why are these necessarily cryptic?

411ff Are these genera names? Should these be in italics?

485 "Viktor" instead of "Victor"

Fig. 3 Genus names should be in italics.

Joachim T. Haug, LMU Munich

Experimental design

ok

Validity of the findings

ok

---

## Round 0.2 · Minor Revisions

The revised submission is much-improved relative to the first submission, but further touch-ups on some of the comments by the reviewer will help to move to accept the publication. Kindly address some of these concerns if possible as soon as possible

Reviewer 1 ·

Basic reporting

Basic reporting
I believe a little language touch up would not come amiss. For instance, I find in line 461 the phrase "known for their high intraspecific variations among species", which is hard to parse. Please also re-read for ease of understanding; e.g., line 479 reads "Therefore, implementing more COI larvae data into studies would help enormously in resolving at least some taxonomic uncertainties".
Raw data possibly supplied on BOLD but cannot be verified.
In Table 3, it would be interesting to see where the alternative «species delimitation» methods differ from RESL and BINs. I would ask the authors to supply this information as well – this is important for gauging how the methods compare and, of course, will be important for future discourse on how to best use «species delimitation» tools in integrative taxonomy.

Experimental design

Adequate.

Validity of the findings

Still valid, yet I am afraid to note that the authors misinterpreted my comment on the Chao1 and Chao2 estimators: my point was that their approach likely underestimated diversity by inflating the probability of encountering a «new» species. I think this should be represented in the manuscript. I do not think that it takes from the main message – that an integrative approach is especially useful when dealing with Chironomidae – but will provide a more realistic estimate of how much diversity there is in the sample. And I think this is important.

Additional comments

Review Chimeno et al. PeerJ 81108 revised version
Custom checks
Deposited data cannot be accessed. DOI link appears to be empty and GenBank accession IDs do not return sequence data (not match but for the Gene database, and there returns a value of 0 – indicating that sequence data is not there)

Basic reporting
I believe a little language touch up would not come amiss. For instance, I find in line 461 the phrase "known for their high intraspecific variations among species", which is hard to parse. Please also re-read for ease of understanding; e.g., line 479 reads "Therefore, implementing more COI larvae data into studies would help enormously in resolving at least some taxonomic uncertainties".
Raw data possibly supplied on BOLD but cannot be verified.
In Table 3, it would be interesting to see where the alternative «species delimitation» methods differ from RESL and BINs. I would ask the authors to supply this information as well – this is important for gauging how the methods compare and, of course, will be important for future discourse on how to best use «species delimitation» tools in integrative taxonomy.

Experimental design.
Adequate.

Validity of the findings
Still valid, yet I am afraid to note that the authors misinterpreted my comment on the Chao1 and Chao2 estimators: my point was that their approach likely underestimated diversity by inflating the probability of encountering a «new» species. I think this should be represented in the manuscript. I do not think that it takes from the main message – that an integrative approach is especially useful when dealing with Chironomidae – but will provide a more realistic estimate of how much diversity there is in the sample. And I think this is important.

Comments en gros
I find the new version much improved, well done. Nonetheless, I am afraid to state that in some aspects the authors did not fully pull through in their implementation of the «species delimitation» methods, e.g., as in Table 3 (see above). Also, there are some linguistic inconsistencies which detract from the overall quality of the manuscript.

Comments en detail
None due to time constraints. My apologies.

Comments on R plotting regarding Figs 3b,c original manuscript
A means of plotting other symbols would simply be to change pch to one of 20:25, and use bg for different colour fills corresponding to line col in the plot command. This may be a bit fiddly in some aspects, but can be achieved if the data is properly structured first.

---

## Round 0.3 · accepted · Accept

Congratulations. After going over the revised manuscript. I find it suitably addresses reviewer comments from previous drafts.